# A Novel Mouse Monoclonal Antibody C42 against C-Terminal Peptide of Alpha-1-Antitrypsin

**DOI:** 10.3390/ijms22042141

**Published:** 2021-02-21

**Authors:** Srinu Tumpara, Elena Korenbaum, Mark Kühnel, Danny Jonigk, Beata Olejnicka, Michael Davids, Tobias Welte, Beatriz Martinez-Delgado, Sabina Janciauskiene

**Affiliations:** 1Department of Respiratory Medicine, Biomedical Research in Endstage and Obstructive Lung Disease Hannover (BREATH), Member of the German Center for Lung Research (DZL), Hannover Medical School, 30625 Hannover, Germany; tumpara.srinu@mh-hannover.de (S.T.); welte.tobias@mh-hannover.de (T.W.); 2Institute for Biophysical Chemistry, Hannover Medical School, 30625 Hannover, Germany; korenbaum.elena@mh-hannover.de; 3Institute of Pathology, Biomedical Research in Endstage and Obstructive Lung Disease Hannover (BREATH), Member of the German Center for Lung Research (DZL), Hannover Medical School, 30625 Hannover, Germany; kuehnel.mark@mh-hannover.de (M.K.); jonigk.danny@mh-hannover.de (D.J.); 4Department of Experimental Medicine, Lund University, SE-221 00 Lund, Sweden; beata.olejnicka@med.lu.se; 5Davids Biotechnologie GmbH, Röntgenstraße 3, 93055 Regensburg, Germany; info@davids-bio.com; 6Department of Molecular Genetics, Institute of Health Carlos III, Center for Biomedical Research in the Network of Rare Diseases (CIBERER), 28220 Majadahonda, Madrid, Spain; 7Department of Genetics and Clinical Immunology, National Institute of Tuberculosis and Lung Diseases, 01-138 Warsaw, Poland

**Keywords:** C-terminal peptide, monoclonal antibody, C42 clone, immunoassays, alpha1-antitrypsin, molecular forms

## Abstract

The C-terminal-fragments of alpha1-antitrypsin (AAT) have been identified and their diverse biological roles have been reported in vitro and in vivo. These findings prompted us to develop a monoclonal antibody that specifically recognizes C-36 peptide (corresponding to residues 359–394) resulting from the protease-associated cleavage of AAT. The C-36-targeting mouse monoclonal Immunoglobulin M (IgM) antibody (containing κ light chains, clone C42) was generated and enzyme-linked immunosorbent assay (ELISA)-tested by Davids Biotechnologie GmbH, Germany. Here, we addressed the effectiveness of the novel C42 antibody in different immunoassay formats, such as dot- and Western blotting, confocal laser microscopy, and flow cytometry. According to the dot-blot results, our novel C42 antibody detects the C-36 peptide at a range of 0.1–0.05 µg and shows no cross-reactivity with native, polymerized, or oxidized forms of full-length AAT, the AAT-elastase complex mixture, as well as with shorter C-terminal fragments of AAT. However, the C42 antibody does not detect denatured peptide in SDS-PAGE/Western blotting assays. On the other hand, our C42 antibody, unconjugated as well as conjugated to DyLight488 fluorophore, when applied for immunofluorescence microscopy and flow cytometry assays, specifically detected the C-36 peptide in human blood cells. Altogether, we demonstrate that our novel C42 antibody successfully recognizes the C-36 peptide of AAT in a number of immunoassays and has potential to become an important tool in AAT-related studies.

## 1. Introduction

The human *SERPINA1* gene is located on chromosome 14q31–32.3 and produces α1-antitrypsin (AAT), a glycoprotein of about 54 kDa, and a prototypical member of the serpin superfamily of proteins. AAT is mainly known as a protease inhibitor forming stable complexes with a range of serine proteases (specifically neutrophil elastase), but also can be targeted by non-specific proteases (e.g., matrix metalloproteases) (MEROPS database: http://merops.sanger.ac.uk/index.shtml (accessed on 17 February 2021)). When AAT forms an inhibitor complex with target proteinases, it is cleaved into large N-terminal and a shorter 36 amino acid C-terminal fragment and becomes irreversibly inactivated. Similar inactivation of AAT also occurs during its cleavage by metalloproteases or cysteine-aspartic proteases [1]. These latter proteases usually cleave AAT at the sites in its reactive site domain, which results in the generation of different variants of C-terminal fragments. For example, a C-44 peptide of AAT has been isolated from human placenta [2] and C-41/42-peptide was detected in nipple aspiration fluids from breast cancer patients [3]. Recently C-36 peptide of AAT was found in plasma of septic patients and has been proposed to be useful as a diagnostic marker [4].

The manifold physiological functions for C-terminal peptides of AAT have also been reported, including suppression of the natural killer cells [5] and control of HIV-1 replication in infected individuals [6]. In models of primary rat hepatocytes and in mice experiments, we previously demonstrated that the C-36 peptide of AAT is a specific down-regulator of the neutral bile acid synthesis pathway [7]. The synthetic C-36 peptide of AAT has also been found to affect proliferation and invasiveness of the human pancreatic adenocarcinoma and breast carcinoma cells [8]. One study provided evidence that the C-terminal peptide of AAT, CSIPPEVKFNKPFVYLI (Y105), interacts with siRNA and can deliver it intracellularly [9].

The fragments resulting from the protease-associated cleavage of AAT might serve as signatures for specific proteases, and therefore, peptidome-degradome profiling of biological fluids might reveal unique AAT peptides with diagnostic and/or prognostic value. In general, such peptides may have broad applications in experimental and clinical research, including the production of anti-peptide antibodies and vaccine development [10]. In the last two decades, many new peptide drugs have been approved for a wide range of conditions, such as cancer, infections, metabolic diseases, and cardiovascular diseases [11]. Altogether, the findings outlined above prompted us to develop a monoclonal antibody that specifically recognizes the C-36 peptide but not a full-length AAT protein and that can be useful to detect the peptide in vitro and/or in biological samples in vivo or can be useful for other translational applications.

## 2. Results

### 2.1. Mouse Monoclonal Antibody (mab) C42 Specifically Recognizes C-36 Peptide in Dot-Blot Assays

Synthetic C-36 peptide was used for the immunization of mice, and hybridoma clones were screened by ELISA. Few positive clones were obtained, but a single clone producing a sufficient amount of monoclonal Immunoglobulin M (IgM) antibody was identified by screening and selected for expansion. The antibody we used after purification since crude culture supernatants usually contain bovine immunoglobulins (www.davids-bio.com (accessed on 17 February 2021)). A purified antibody (hereafter referred as C42 mAb) was therefore used for further analyses. To assess the specificity, we tested C42 mAb using AAT protein and synthetic C-36 peptide by dot-blot analysis. As illustrated in Figure 1A, C42 reacts with C-36 peptide and does not react with a full-length AAT protein. Based on a dot-blot assay, we concluded that the lower amount of C-36 peptide recognized by C42 mAb ranges between 0.2–0.05 µg.

We next investigated whether C42 mAb is cross-reacting with other known molecular forms of AAT, such as oxidized, polymeric, and complexed with elastase [12,13]. As shown in Figure 2, none of the tested molecular forms of AAT were detected with C42 mAb in dot-blot assays. To further access antibody specificity, we also evaluated C42 mAb reactivity with shorter C-terminal peptides of AAT, such as C-10 (H-AGAMFLEAIP-OH), C-21 (H-MIEQNTKSPLFMGKWNPTQK-OH), and Y105 (H-SIPPEVKFNKPFVFL-OH). According to our dot-blot assays, none of the tested fragments were recognized by C42 mAb (Figure 2).

### 2.2. C42 mAb Does Not Recognize Denatured C-36 Peptide after Electrophoresis Following Western Blotting

In dot-blot assays, the C-36 peptide was not denatured but was spotted through circular templates directly onto the membrane. To validate whether C42 mAb can detect denatured C-36 peptide after electrophoretic separation we employed SDS-PAGE followed by semi-dry Western blotting. The C-36 peptide is cationic and small (pI/Mw: 9.82/4134.95) and, therefore, required optimization of the electrophoresis and Western blotting conditions. We also optimized the blocking conditions using 5% fat-free milk or 5% bovine serum albumin, among others. We were able to confirm that C-36 peptide samples boiled to denature the protein prior to 16.5% Tricine/Tris SDS-PAGE and followed by Western blotting could be detected by the rabbit polyclonal anti-human AAT antibody (Figure 3). Unfortunately, under the same experimental conditions, C42 mAb did not react with denatured C-36. Thus, C42 mAB recognizes C-36 peptide in its native sate.

### 2.3. Localisation of the C36 Peptide in Human Naive and Lipopolysaccharide (LPS)-Treated Neutrophils

We previously reported that human neutrophils express and release C-terminal peptide(s) of AAT [14]. For visualization of the intracellular C36 peptide in freshly isolated human blood neutrophils, we employed C42 mAb in laser scanning confocal microscopy. It is well known that an antibody binds differently to its antigen dependent on the various fixation substances, and therefore, we tried several cell fixation methods that could be suitable for our new C42 mAb. To preserve cellular structures in their native conformation as much as possible, we fixed cells with paraformaldehyde and further permeabilized with 0.5% Triton X-100. As illustrated in Figure 4A, all neutrophils were positive for the C-36 peptide with a signal accumulated in granular-like structures throughout the cell. The negative control experiment with non-reactive IgM (isotype control) showed very weak background signal, thus confirming specificity of the C42-immunolabeling (Figure 4A). To trigger activation and extracellular traps (NETs) formation, we incubated neutrophils in medium supplemented with 20 ng/mL LPS for 2 h prior to fixation. The C36-positive signal was detected in the NETs structures although it did not localize with DNA fibers released from activated neutrophils (Figure 4B).

To explore whether C-36 peptide co-localizes with a full-length AAT protein in neutrophils, we performed double-labeling experiments. Confocal microscopy revealed that the distribution patterns of C-36 peptide and a full-length AAT were largely different with only a few overlapping spots (Figure 5).

Even less overlap was observed between C-36 and a full-length of AAT in neutrophils releasing DNA upon LPS stimulation (Figure 6). The C-36 peptide was predominantly localized around and within the nuclei, whereas AAT was strongly associated with NETs and located in the cytosol. Analysis of the orthogonal sections of 3D scanned specimens confirmed that C-36 and AAT only slightly co-localize in neutrophil cytosol but show almost no co-localization in the NETs structures (Appendix A). This confirms that C42 mAb is highly specific for the C-36 peptide and distinguishes it from a full-length AAT protein.

### 2.4. Assessment of DyLight Labelled C42 mAb Suitability for Flow Cytometry (FACS) Analysis of Human Neutrophils

We next employed C42 mAb labelled with DyLight^®^ 488 NHS ester for the flow cytometry (FACS) analysis. Freshly isolated neutrophils (1.5 × 10^6^ cells per condition) were incubated in non-adherent plates for 2 h and stained with DyLight488 conjugated C42 mAb. As shown in Figure 7, most of the neutrophils were positive for C-36 peptide under basal conditions. Because C-terminal peptides have been demonstrated to enter (or interact) with cells [6,9], we performed parallel experiments in which freshly isolated neutrophils were incubated for 2 h in the presence of synthetic C-36 peptide (1 μg/mL) and afterwards stained with DyLight488- C42 mAb. As illustrated in Figure 7, staining significantly increased in cells pre-treated with the C-36 peptide as compared to those non-treated with C-36 peptide. These results confirm findings presented in Figure 5 and Figure 6 and suggest that C42 mAb can be applied for the FACS analyses.

### 2.5. Intracellular Localisation of the C36 Peptide in Human Total Peripheral Blood Mononuclear Cells (PBMCs)

Finally, we used C42 mAb to analyze the localization of C-36 peptide in total adherent human peripheral blood mononuclear cells (PBMCs). Immunolabeling revealed a number of C-36-positive cells (Figure 8A). However, the signal intensity and subcellular distribution pattern varied significantly between cells. Negative controls with primary antibodies omitted or substituted for non-reactive IgM confirmed the specificity of the C42 mAb (Figure 8A). To assess the relative distribution of the C-36 peptide and the full-length AAT protein in adherent PBMCs, we performed double labeling experiments with C42 mAb and rabbit polyclonal anti-AAT antibodies. Three-dimensional reconstruction of the confocal sections revealed partial co-localization between C-36 and AAT protein-signals. The full-length AAT protein we detected pre-dominantly at the cell periphery whereas C-36 peptide was mostly present in the perinuclear areas (Figure 8B). These different intracellular localization patterns imply that AAT and C-36 peptide may have putatively distinct functions, and requires further ivestigations.

## 3. Discussion

It is well recognized that C-terminal peptides of AAT have diverse biological activities and might be associated with a multitude of pathophysiological conditions. The size of these peptides is typically about 36–42 amino acids, believed to result from the target or non-target protease cleavages within the reactive center loop of AAT molecule. Those points have recently been challenged by new findings that human cells may express not only a full-length AAT protein but also 358 bp (ST1C4) and 248 bp (ST1C5) short transcripts of, and that these transcripts have an open reading frame in exon V and, thus, can produce peptides [14]. This finding gave a novel insight into the transcription of SERPINA1 gene, suggesting that short peptides of AAT, such as C-36, can derive from the cleavage of full-length AAT but putatively may also be expressed by human cells [14].

Several studies have found that C-terminal peptides generated from the reactive center of AAT but also from other serpins, such as the mammalian serpin neuroserpin and myxomaviral serpin, express potent immune modulatory activities [15,16]. It has been postulated that serpin-derived peptides may have numerous applications in research as well as in clinical diagnostics, treatments, and/or vaccine development. We, therefore, used synthetic C-36 peptide of AAT to generate a peptide-specific antibody as a putative tool for immunological research approaches.

In collaboration with Davids Biotechnologie GmbH Company, we established a hybridoma cell line that stably produces anti-C36 peptide mouse monoclonal antibodies named as IgM C42 mAb. Notably, two independent efforts to obtain a mouse monoclonal immunoglobulin G (IgG) anti-C-36 antibody were unsuccessful, allowing us to conclude that C-peptide of AAT generates an immune response characterized by the persistent production of antibodies of the IgM class. The IgM antibody is more difficult to work with in terms of reagent availability, purification, and specificity; additionally, the antigen binding affinities of IgM antibodies are typically lower than IgG [17]. On the other hand, the pentameric structure of IgM renders them particularly efficient at binding antigens present at low levels more strongly than monomeric IgGs [18].

The specificity of C42 mAb was confirmed by dot-blot analyses. The C42 mAb reacted well with C-36 peptide but showed no cross-reactivity with native, polymeric, or oxidized full-length AAT protein, as well as with the complex mixture of AAT with elastase or shorter C-terminal peptides of AAT. The actual epitope that is recognized by the C42 antibody is unknown, but it became unavailable after peptide denaturation and separation by electrophoresis in polyacrylamide gels following blotting. Different attempts to change antibody dilutions, blocking times, and/or concentrations of blocking agents, which can mask antibody-antigen interactions, failed to show the suitability of the C42 mAb to detect denatured C-36 peptide in Western blotting assays. The important to mention that, under the same experimental conditions, rabbit polyclonal anti-AAT antibody detected the C-36 peptide in Western blots, confirming that the peptide was transferred from the blotted membranes. Hence, it appears that after denaturation, the C42 mAb binding epitope on the C-36 peptide is lost. We know that non-denaturing PAGE, also called native-PAGE, separates proteins according to their mass/charge ratio. However, since our short peptide is highly cationic, we were not able to use the Laemmli system with pH 8.8 and were unsuccessful with native-PAGE/Western blotting assays. Alternative gel types, such as agarose gels, may be better suited for evaluating our antibody in western blots [19]. Many antibodies towards conformational epitopes did not bind their target proteins in the Western blot assays [20]. Thus, C42 mAb probably are conformational or spatial structure-dependent. Because conformational epitopes are difficult to map with linear peptides, the cyclic constrained peptides mimicking looped epitope structures will be employed for further studies. Detailed information related to the C42 epitope will assist us to improve antibody applications and will provide better understanding about C-36 peptide–antibody interactions.

We next tested the suitability of C42 mAb for other applications, such as confocal laser scanning microscopy and flow cytometry. Based on our previous findings that C-36 peptide is present and secreted by human neutrophils [14], we used C42 mAb to stain freshly isolated human blood neutrophils under the basal conditions or when neutrophils were pre-activated with LPS. The specific C36-positive signal was observed in granular-like structures throughout the cells. In LPS-activated neutrophils, C36-immunoreactivity was detected in networks of extracellular traps (NETs), structures typically consisting of de-condensed DNA associated with histones and antimicrobial peptides derived from the granular and nuclear contents of neutrophils [21]. Remarkably, C-36 peptide does not co-localize with DNA fibers, and thus, further investigations are of interest to clarify the relationship between NETosis and C-36 peptide.

A previous study by Clemmensen et al. [22] found that AAT is a constituent of neutrophil granules. Therefore, we performed double labeling experiments using mouse C42 mAb and rabbit polyclonal anti-AAT antibodies to investigate the distribution of the C36 peptide and full-length AAT in human neutrophils. Interestingly, we found very limited co-localization between C-36 and a full-length AAT in untreated as well as in LPS-stimulated neutrophils. This finding does not exclude that C-36 peptide may be generated during the cleavage of AAT, but still allows us to consider the possibility that C-36 is expressed by neutrophils, as proposed in our previous study [14].

Labeling of antibodies is used for fluorescence detection methods; therefore, we conjugated C42 mAb with DyLight^®^ 488 NHS ester to evaluate its suitability in flow cytometry assays. In line with results from confocal laser scanning microscopy, most of the neutrophils were positive for the C-36 peptide under basal conditions, whereas staining significantly increased in cells pre-treated with synthetic C-36 peptide. Knowledge gained from the above experiments shows that C42 mAb could be applied to detect C-36 peptide in human blood neutrophils and possibly in other cells and/or biological samples.

Indeed, we performed double labeling experiments to check for the distribution of the C36 peptide and the full-length AAT protein in adherent total human PBMCs. The 3D reconstruction of the confocal sections revealed only partial co-localization of the C-36 and AAT-signals in PBMCs. The full-length AAT pattern was equally strong on the cell surface and intracellular structures, whereas C-36 peptide was predominantly found in the cytosol, implying putative differences in intracellular functions between AAT and its peptide. Future work will focus on uncovering the pathways involved in the generation of C-36 peptide and its functions in human immune cells.

Hence, unconjugated and conjugated C42 mAb can be useful tools for immune histochemistry and flow cytometry analyses of C-36 peptide in human blood cells during health and disease. The identification and/or quantification of native AAT and its conformers are critical for a fundamental understanding of the biological role of AAT. Post-translational modifications of AAT expand its chemical and functional properties and play key roles in many biological processes, such as cell signaling and regulation of gene transcription, and are associated with liver, lung, and other diseases.

The conformational changes that occur in the AAT protein during post-translational modifications lead to the generation of conformation-dependent neoepitopes [23]. However, our current reports and many others show that the generation of highly specific antibodies against conformational epitopes by animal immunization remains challenging [24]. Previously monoclonal antibody 3F4 was generated against oxidized AAT that did not react with either the native AAT or the elastase-AAT complex [25]. Moreover, few monoclonal antibodies (2C1 and ATZ11) that recognize a conformation-dependent neoepitope on polymerized forms of native and/or genetic mutants of AAT have been put forward [26,27]. Nevertheless, these conformation-specific antibodies have limitations in immunogenicity and their applications. In general, there is an increasing number of new antibody libraries designed specifically toward post-translationally modified proteins. Therefore, careful validation of monoclonal antibodies, like C42 mAb, can serve as a useful resource that potentially accelerates the generation of better-quality antibodies against AAT fragments, which may in turn broadly benefit the scientific community [28].

Taken together, we present a novel IgM C42 mAb that recognizes C-36 peptide of AAT but that has no affinity for the AAT protein itself. Based on C42 mAb immunoassays, we also confirm that a fraction of AAT in immune cells exists in a peptide form. In order to test the suitability of C42 mAb for further applications, we performed an enzyme-linked immunosorbent assay (ELISA) in which the concentration-dependent capture of C42 mAb by the immobilized C-36 peptide on the surface of microplate wells was tested (data not shown). Based on our promising results, additional research (currently under way in our laboratory) is being done to test this approach in terms of the best concentration of antibody, the best concentration of C-36 peptide, the suitability of biological samples for the assay, and so on. If successful, the development of this assay would be a step forward into the quantification of C-36 peptide and might be of assistance in developing new, antibody based, methods for the detection of AAT peptides in biological samples.

Hence, our C42 antibody can be a valuable tool for the qualitative and quantitative studies of C-terminal peptide of AAT. More broadly, our results further highlight the monoclonal antibody as an essential implement for detecting post-translationally modified molecular forms of human proteins in health and diseases.

## 4. Materials and Methods

### 4.1. C-Terminal Peptides of AAT Protein

High quality peptides of AAT C-10 (H-AGAMFLEAIP-OH), C-21 (H-MIEQNTKSPLFMGKWNPTQK-OH), Y105 (H-SIPPEVKFNKPFVFL-OH), and C-36 (H SIPPEVKFNKPFVFLMIEQNTKSPLFMGKWNPTQK-OH) (above 95% according to high performance liquid chromatography and mass spectrometry analyses) were obtained from JPT Peptide Technologies GmbH, Berlin, Germany.

### 4.2. Native AAT Protein

Human plasma purified AAT (20 mg/mL, 99% purity, Zemaira, CSL Behring, Marburg, Germany) was used for experiments after buffer exchanged to the sterile phosphate buffered saline (PBS), (Gibco Life technologies, Carlsbad, CA, USA) by using 10 K centrifugal filter columns (Thermo Fischer Scientific, MA United States).

### 4.3. Preparation of Oxidized AAT

Oxidized AAT was prepared by adding N-chlorosuccinimide (Sigma, St. Louis, MO, USA) to native AAT dissolved in PBS at 25:1 molar ratio. The mixture was allowed to react for 30 min at RT, and N-chlorosuccinimide was then removed by washing and equilibrated with PBS using viva spin-20 centrifugal columns (Sartorius, Gottingen, Germany). Oxidized AAT did not form complex with elastase as it has been shown in our previous studies [29].

### 4.4. Preparation of Polymerized AAT

The AAT protein was diluted with PBS, pH 7.4, to generate a 2 mg/mL AAT stock solution and was heated at a fixed temperature of 60 °C for 3 h. Afterwards, protein polymers were confirmed by using 7.5% SDS-PAGE without sample heating and by 7.5% native PAGE.

### 4.5. Preparation of AAT-Pancreatic Elastase Complex

Native AAT and elastase from porcine pancreas (Sigma-Aldrich, St. Louis, MO, USA) were mixed at 1.5:1 molar ratio (AAT/elastase) and incubated for 30 min at RT as previously described [30]. The complex mixtures were used immediately.

### 4.6. Generation of Anti-C36 Peptide Mouse Monoclonal Antibody

Synthesized C-36 peptide (SIPPEVKFNKPFVFLMIEQNTKSPLFMGKVVNPTQK), 78% purity, was conjugated to a carrier before immunization and mouse monoclonal antibody against C-36 peptide was produced, validated, and purified by Davids Biotechnologie GmbH, Germany according to the protocols established by the company (https://www.davids-bio.com/pages/antibodies.html (accessed on 17 February 2021)). An immunization protocol of three mice of specific strain (*MAR-SIP*) was following immunizations with 40 µg antigen on days 1, 14, 28, 35, 42, 56, 87 and 97. Test bleedings occurred at day 1 (pre-immune), and at day 35 (immune). To determine the concentration of specific anti-C-36 antibody in antisera, the ELISA assay, developed by Davids Biotechnologie GmbH, was used. In brief, ELISA plates were coated with C-36 peptide, prepared in 0.1 M bicarbonate buffer, pH 9, and incubated for 24 h at 4 °C. Afterwards, plates were washed, blocked for 1 h at 26 °C, and different dilutions of antiserum were added for 24 h at 4 °C. After, plates were washed, and alkaline phosphatase conjugated antibody was added for 4 h at 26 °C. The reaction was stopped with 2N NaOH and analyzed in ELISA reader at 405 nm.

The animal with the best immune response received a final immunization to boost the antibody producing cells. Splenic cells were isolated for direct fusion with immortal myeloma cell lines. Sub-cloning with the limited-dilution method was performed, which ensured that the IgM C42 clone is truly monoclonal, having the long-term stability. For the identification of mouse monoclonal antibody isotypes, subtypes, and light chains, we used Milenia^®^ QuickLine Mouse Immunoglobulins-Universal module (MQUM 1) (Milenia Biotec GmbH, Germany) according to kit recommendations. For antibody purification, the company employed the specific methods. Stock concentrations of purified C42 mAb were 0.75 and 1.17 mg/mL.

### 4.7. Dot-Blots

Different amounts of C-36 peptide and a constant amount (5 µg) of different molecular forms of AAT (native, polymerized, oxidized, and elastase complex) were loaded into the 0.2 µm nitrocellulose membranes (GE healthcare life sciences, Freiburg, Germany). After drying for at least 2 h, at RT, membranes were blocked with Tris-buffered saline (TBST) containing 0.1% Tween^®^ 20 and 5% low fat milk for 1 h at RT and incubated overnight with rabbit polyclonal anti-human AAT antibody (1:800, DAKO, Glostrup, Denmark) or mouse IgM C42 mAb (1:500, Davids Biotechnologie, Germany). In some experiments we used mouse monoclonal anti-AAT antibody IgG B9 (1:1000, Santa Cruz, TX, USA).The immune complexes were visualized with suitable horseradish peroxidase-conjugated secondary antibodies (anti-rabbit or anti-mouse 1:10,000, DAKO, Glostrup, Denmark) and enhanced by ECL Western blotting substrate (Bio-Rad, München, Germany). Images were taken using Chemidoc Touch imaging system (Bio-Rad, Hercules, CA, USA).

### 4.8. Dot-Blots Confirming that C42 mAB Does Not Bind to the Carrier Protein of the C-36 Peptide

We pre-incubated equal volumes of C42 mAb (10 µg/mL) with C-36 peptide (5 µg/mL, JPT Peptide Technologies GmbH, Berlin, Germany) for 1 h, at RT. Afterwards, C42 mAb alone or pre-incubated with C-36 peptide was used for dot-blot assay as above. As shown in Figure 9, antibody pre-incubated with the peptide strongly lost ability to detect C-36 peptide loaded into the 0.2 µm nitrocellulose membranes. When 10-fold excess of the C-36 was used for the pre-incubation with C42 mAb, antibody does not detect C-36 peptide in the dot-blots (data not shown).

### 4.9. Western Blots

The full-length AAT protein and C-36 peptide were heated at 90 °C for 3 min and separated by 16.5% SDS-PAGE (Bio-Rad, Hercules, CA, USA) prior to transfer onto 0.2 µm polyvinylidene difluoride (PVDF) membranes (SERVA, Heidelberg, Germany). Membranes were blocked for 1 h with TBS + 0.01% Tween-20 containing 5% low fat milk powder (Roth, Karlsruhe, Germany), followed by overnight incubation with primary antibody anti-human AAT (1:800, rabbit polyclonal, IgG, DAKO) or mouse IgM C42 mAb (1:500, Davids Biotechnologie, Germany) at 4 °C. The immune complexes were visualized with suitable horseradish peroxidase-conjugated secondary antibodies (anti-rabbit or anti-mouse 1: 10,000, DAKO) and enhanced by ECL Western blotting substrate (Bio-Rad). Images were taken using Chemidoc Touch imaging system (Bio-Rad).

### 4.10. Human Blood Neutrophil Isolation

Human neutrophils were isolated from fresh peripheral blood of healthy volunteers using polymorphprep (Axis-Shield PoC AS, Oslo, Norway) according to the manufacturer’s recommendations as elsewhere described [31].

### 4.11. Human Blood PBMCs Isolation

Total PBMCs were isolated from peripheral blood using Lymphosep (C.C Pro, Oberdorla, Germany) discontinuous gradient centrifugation according to the manufacturer’s instructions as previously described [32]. PBMCs were then re-suspended in serum free RPMI 1640, L-glutamine supplemented medium (Gibco Lifetechnologies, Carlsbad, CA, USA) and plated on glass coverslips at a density of (2 × 10^6^) per condition. Cells were allowed to adhere to the glass surfaces for 2 h at 37 °C and 5% CO_2_ [33] before experiments.

### 4.12. Evaluation of Isolated Blood Cells by Cytospins

Immediately after the isolation, cells (neutrophils or PMBCs) were suspended in RPMI media to a concentration of approximately 0.5 × 10^6^ cells/mL; 250 µl of cell suspension was added to a slide chamber and centrifuged in cytocentrifuge (Cellspin-I, Tharmac GmbH, Wiesbaden, Germany) at 500 rpm for 5 min. Afterwards, slides were air dried prior to proceeding with staining procedure. Slides were stained with May-Grünwald (Merck Millipore, Darmstadt, Germany) for 3–5 min followed by washing in deionized water for 3 min, and staining with Giemsa solution for 5 min. Finally, slides were rinsed in deionized water, air dried, and evaluated microscopically (Leica DM 750, LEICA, Wetzlar, Germany) (Figure 10)

### 4.13. Monoclonal C42 Antibody Labeling with DyLight 488

C42 mAb conjugation was performed by using Dylight^®^ 488 conjugation kit (Abcam, Cambridge, UK) according to the manufacturer’s recommendations. Briefly, 10 µL of modifier and 100 µL (160 µg) of C42 antibody were added into conjugate reaction tube, mixed gently by pipetting, and incubated for 30 min at RT. Conjugation reaction was terminated by adding 10 µL of quenching solution for 10 min at RT. To the conjugated antibody, 0.1% NaN_3_ was added and stored at 4 °C until use. The aggregates were removed from antibody solution by gentle centrifugation (300 g for 3 min) before starting our flow cytometry experiments.

### 4.14. Immunofluorescence Confocal Laser Microscopy

Human neutrophils (1.5 × 10^6^) or human PBMCs (2 × 10^6^) were plated onto glass coverslips and incubated in RPMI medium for 2 h at 37 °C and 5% CO_2_. For the stimulation experiments, neutrophils were incubated in RPMI supplemented with 20 ng/mL LPS (Sigma Aldrich, Darmstadt, Germany). Cells were then washed with PBS, fixed with 3% PFA/PBS (paraformaldehyde/phosphate buffered saline) for 20 min and permeabilized with 0.5% Triton X-100 in PBS for 5 min at RT. For immunolabeling, cells were incubated with primary antibodies targeting human AAT (1:5000, rabbit polyclonal, IgG, DAKO) and C-36 peptide (1:500, IgM, clone C42 mAb, Davids Biotechnologie) separately or in combination for 1 h, at RT. For isotope control, the non-reactive mouse IgM, clone11E10 (Thermo Fisher Scientific, Carlsbad, CA, USA) were taken instead of specific IgM C42 mAb. After washing, the cells were incubated with corresponding secondary antibodies conjugated to AlexaFluor-488 (goat anti-rabbit, IgG (H + L) cross adsorbed antibody, Thermo Fischer Scientific, Rockford, USA) or Dylight-594 (goat anti-mouse, IgM cross adsorbed antibody, Thermo Fisher Scientific, Rockford, USA). In experiments where C42mAb was labelled with DyLight-488 conjugated C42 antibody, secondary antibodies were omitted. After final washing in PBS, the cells were mounted on microscope slides using ProLong Gold Antifade Mountant with DAPI (Thermo Fisher Scientific, Carlsbad, CA, USA). Images were acquired using confocal laser microscope Olympus FluorView 1000 equipped with a 60x oil immersion objective in sequential mode. Confocal z-stacks were collected with a 0.25 µm increment.

### 4.15. Analysis of Neutrophil C-36 by Flow Cytometry

Neutrophils (1.5 × 10^6^ cells) were plated in multi-well suspension culture plates with cell-repellent surface (Greiner bio-one Kremsmünster, Upper Austria), and incubated for 2 h without or with 1 μg/mL C-36 peptide (used as a positive control). Cells were then collected into FACS tubes, washed with PBS (500 *g*, 5 min), and fixed with 3% PFA (paraformaldehyde) for 15 min at RT. Afterwards, cells were washed 3 times with PBS + 2% FCS and permeabilized with 0.2% Triton X-100 or 15 min at RT. Staining was performed with Dylight-488 conjugated C42 mAb. Samples were measured on Guava easyCyte flow cytometer (TX, USA) and data analyzed with FlowJo v10 (NJ, USA). Unlabeled cells were used as controls to set the flow-cytometry parameters.

### 4.16. Statistics

Student’s *t*-test was applied to compare two sample means on one variable. Data were presented as mean (SD). A *p*-value of less than 0.05 was considered significant. Statistics and data visualization were performed using Sigma Plot 14.5 software package (San Jose, CA 95,131 USA).

## Figures and Tables

**Figure 1 ijms-22-02141-f001:**
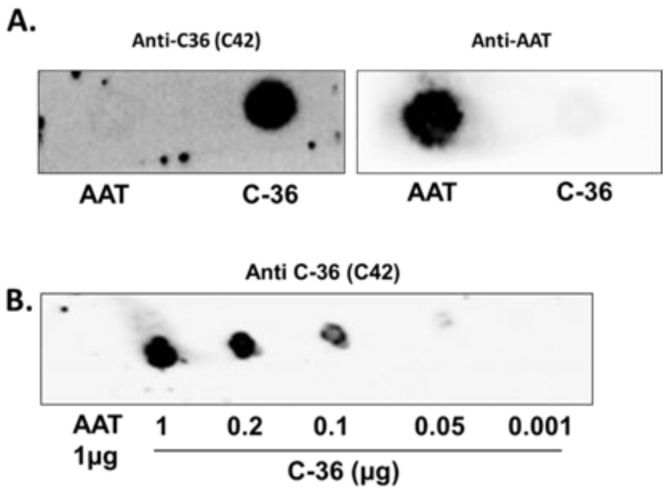
The specificity of the purified antibody (C42 mAb) tested by dot-blot analysis. (**A**). Full-length plasma purified alpha1-antitrypsin (AAT) (1 µg) and C-36 peptide (1 µg) or (**B**) various amounts of C-36 peptide were blotted onto the nitrocellulose membrane and the dot-blotted membranes were incubated with a mouse C42 mAb (1:500). In (A), the same blot we developed again with mouse monoclonal anti-human AAT antibody (1:1000). The immune complexes were visualized with horseradish peroxidase-conjugated secondary anti-mouse antibody (1:10,000) and enhanced by ECL substrate. Images created using Chemidoc Touch imaging system. Representative dot-blots show that (**A**) C42 mAb reacts with C-36 peptide of AAT but not with AAT protein and (**B**) C42 mAb recognizes C-36 peptide in a concentration dependent manner.

**Figure 2 ijms-22-02141-f002:**
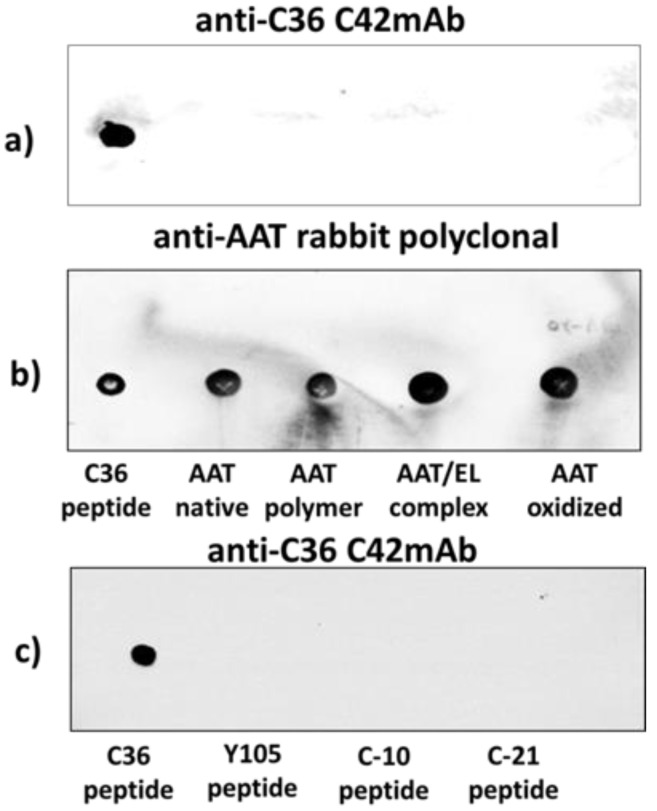
The C42 mAb does not react with different molecular forms of AAT protein and shorter C-terminal peptides of C-36 in dot-blot analyses. (**a**,**b**). Full-length native, polymeric, complexed with elastase, oxidised AAT (all at 5 µg) and C-36 peptide (1 µg) were applied into the nitrocellulose membrane. The dot-blotted membranes were probed with C42 mAb (1:500) and after re-probed with a rabbit polyclonal antibody anti-human AAT (1:800). The immune complexes were visualized with appropriate horseradish peroxidase-conjugated secondary antibodies (1:10,000) and enhanced by ECL blotting substrate. Images created with Chemidoc Touch imaging system. Representative image shows that C42 mAb specifically reacts only with the C-36 peptide. (**c**) Representative image shows that C42 mAb does not react with three shorter fragments of the C-36 peptide (Y105, C-10, and C-21 (all loaded at 1 µg)) but reacts with C-36 (1 µg).

**Figure 3 ijms-22-02141-f003:**
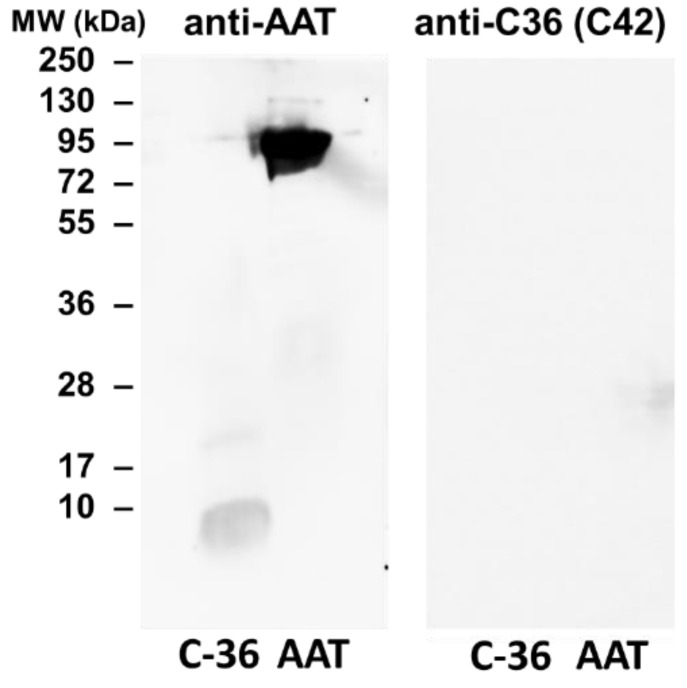
A full-length AAT protein and C-36 peptide separated by 16.5% SDS-PAGE gels, followed by Western blotting using polyclonal rabbit anti-human AAT (anti-AAT) and mouse monoclonal anti-C-36 peptide antibody (C42). Protein samples prepared for SDS-PAGE analysis were denatured by heating at 90 °C for 3 min. After electrophoretic separation, proteins were transferred into 0.2 µm polyvinylidene difluoride (PVDF) membranes. Membranes were blocked for 1 h with TBS+ 0.01% Tween-20 containing 5% low fat milk powder and incubated overnight at 4 °C with rabbit polyclonal anti-human AAT (1:800) or C42 mAb (1:500). The immune complexes were visualized with appropriate horseradish peroxidase-conjugated secondary antibodies and enhanced by ECL Western blotting substrate. Images obtained with Chemidoc Touch imaging system. Representative blot shows that polyclonal anti-AAT antibody reacts with both -native AAT and C-36 peptide, whereas under the same experimental conditions, C42 mAb does not recognize proteins.

**Figure 4 ijms-22-02141-f004:**
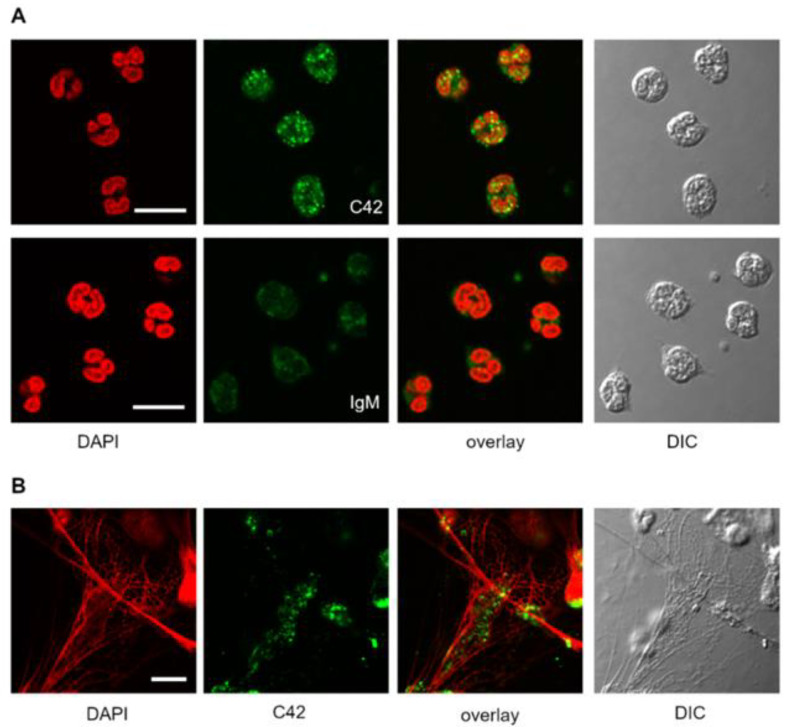
Localization of the C36 peptide of AAT in human neutrophils. (**A**) C36-positive signal (*green*) was observed in granular-like structures throughout the cells. Lack of the signal in the isotype control with non-reactive IgM (lower raw) confirmed specificity of the C42 mAb. (**B**) C36-immunoreactivity (*green*, stained with C42 mAb) was detected in extracellular traps (NETs) structures, albeit without co-localization with DNA fibers (*red*). Nuclei and DNA fibers were defined by 4′,6-diamidino-2-phenylindole (DAPI, *red*). Scale bar, 10 µm.

**Figure 5 ijms-22-02141-f005:**
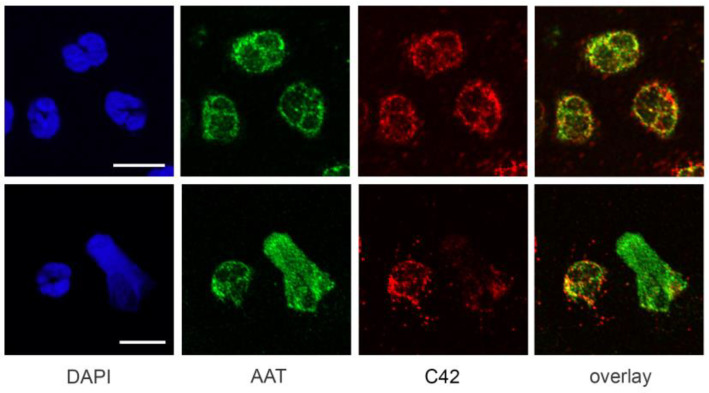
Distribution of the C36 peptide and full-length AAT in human neutrophils. Double labeling experiment showed very limited co-localization between C-36 (*red*, stained with C42 mAb) and full-length AAT (*green*, stained with polyclonal anti-AAT) patterns in untreated (upper raw) and LPS-stimulated (lower raw) neutrophils. Nuclei and DNA fibers were stained by DAPI (*blue*). Scale bar, 10 µm.

**Figure 6 ijms-22-02141-f006:**
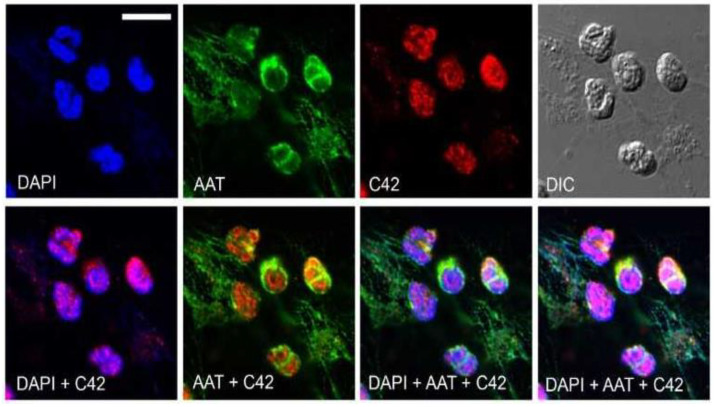
Distribution of the C36 peptide and full-length AAT in NETs from human neutrophils. Double labeling experiment showed very limited co-localization between full length AAT (*green*, stained with polyclonal anti-AAT) and C-36 (*red*, stained with C42 mAb) patterns in LPS-stimulated neutrophils releasing DNA fibers (NETs). Nuclei and DNA fibers stained with DAPI (*blue*). Two last panels marked as DAPI +AAT +C42, show the same image, but second one is overexposed in blue and red channels to better show DAPI staining and C-36 localization in fine DNA fibers. Scale bar, 10 µm.

**Figure 7 ijms-22-02141-f007:**
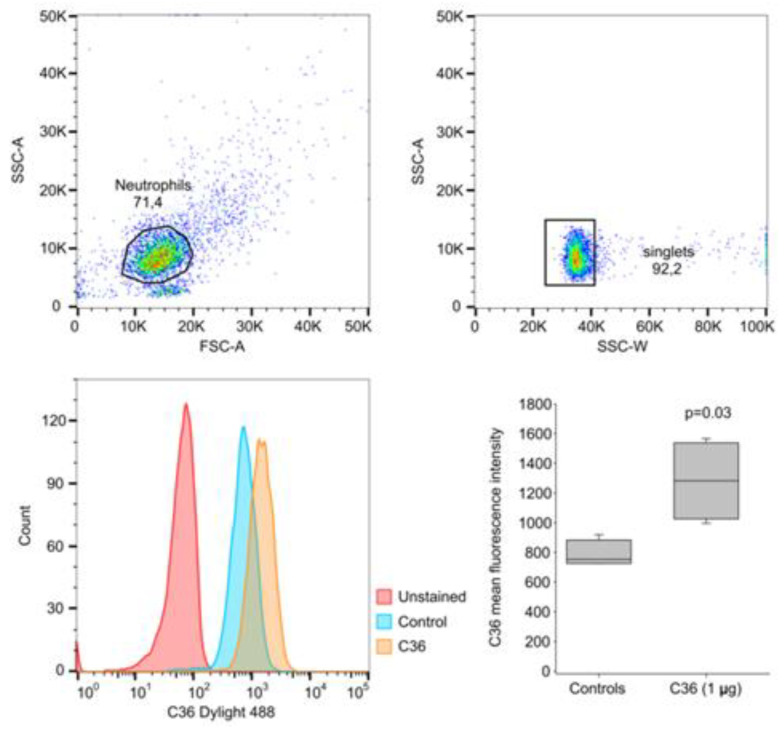
Flow cytometry analysis of C-36 peptide in human neutrophils. Neutrophils (1.5 × 10^6^ cells per condition) were incubated with or without C-36 peptide (1 μg/mL), for 2 h. Cells were then collected and processed as described in materials and methods. Staining was performed with DyLight488 conjugated C42 mAb. Samples were measured with Guava EasyCyte flow cytometer, and data analyzed with FlowJo v10. Unlabeled cells were used as controls to set the flow-cytometry parameters. After incubation with C-36, neutrophils show significantly increased positive staining for the C-36 in comparison to untreated cells. Box and whisker plots lines represent means (SD) of four independent experiments. P was calculated based on the Shapiro–Wilk normality test.

**Figure 8 ijms-22-02141-f008:**
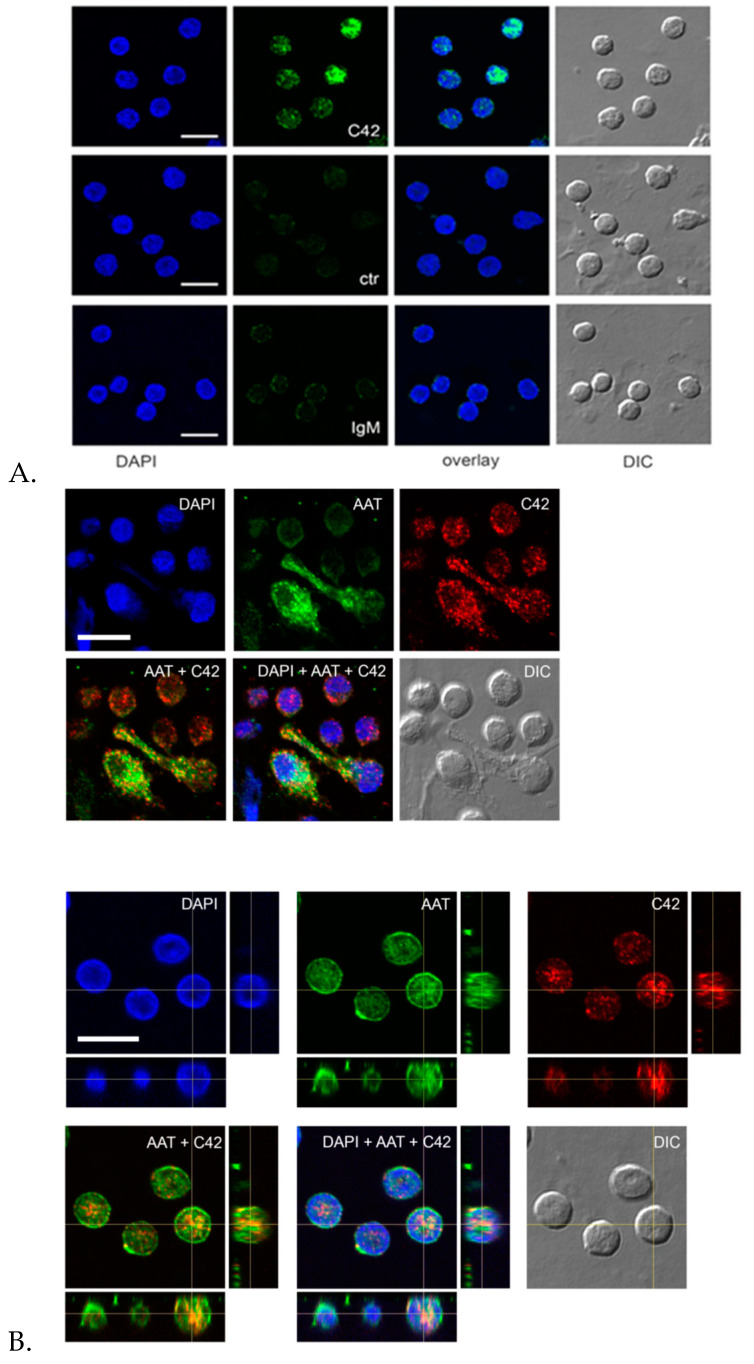
Distribution of the C36 peptide and full-length AAT in human peripheral blood mononuclear cells (PBMCs). (**A**) Immunolabeling with C42 mAb revealed different localization patterns of varying intensity (*green*) in PBMCs (upper raw). Negative controls with primary antibodies omitted (Figure 8A middle raw) or substituted for non-reactive IgM control (Figure 8A, lower raw) showed no or very week fluorescence in comparison to the C42 mAb-signal (Figure 8A, *green* color, upper raw). (**B**) Double labeling of human PBMCs with C42 mAb and anti-human AAT (AAT) revealed distinct, albeit partially co-localizing, patterns of C36 (*red*) and full-length AAT (*green*) signals. Analysis of the orthogonal sections of 3D scanned specimen revealed that C36 was predominantly enriched in nuclei, whereas AAT was present throughout the cell with enhanced labeling of cell periphery. Nuclei and DNA fibers were stained by DAPI (4′,6-diamidino-2-phenylindole, blue). DIC, differential interference contrast. Scale bar, 10 µm.

**Figure 9 ijms-22-02141-f009:**
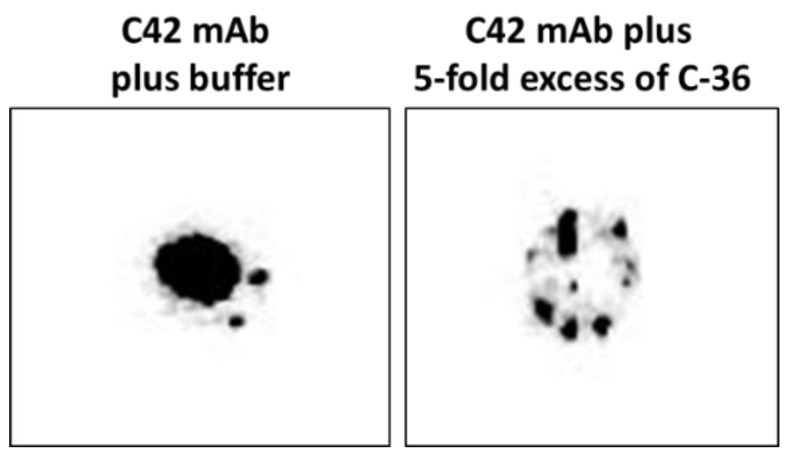
The C42 mAb pre-incubated with about five times excess of the C-36 peptide loses ability to react with C-36 in dot-blot analyses. We applied C-36 peptide (1 µg) into the nitrocellulose membrane. The dot-blotted membranes we probed with C42 mAb alone or with C42 mAb pre-incubated with C-36 peptide (1:800). The immune complexes were visualized with appropriate horseradish peroxidase-conjugated secondary antibodies (1:10,000) and enhanced by ECL blotting substrate. Images created with Chemidoc Touch imaging system. The representative image shows that C42 mAb well recognizes the C-36 peptide, whereas the pre-incubation of the antibodies with C-36 peptide for 1 h, eliminates this reactivity.

**Figure 10 ijms-22-02141-f010:**
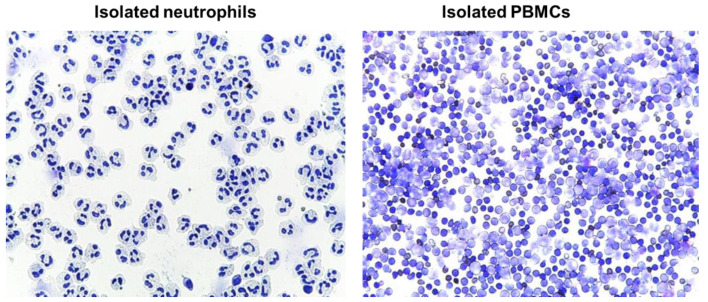
Representative cytospins from isolated human blood neutrophils (*left panel*) and total PBMCs (*right panel*). Images taken at 40× magnification.

## Data Availability

Not applicable.

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
