# Peer review of "A Novel Mouse Monoclonal Antibody C42 against C-Terminal Peptide of Alpha-1-Antitrypsin"

_ijms, 2021, doi:10.3390/ijms22042141_

Round 1

Reviewer 1 Report

In the manuscript entitled “A novel mouse monoclonal antibody C42 against C-terminal 2 peptide of alpha-1-antitrypsin”, Tumpara et al aims to examine the specificity of a novel mouse monoclonal antibody (clone C42) which was generated by Davids Biotechnologie GmbH by dot-blot.  For this purpose, the authors performed several immunoassays including dot-blot and Western blotting.  The dot-blot results indicated that the monoclonal antibody specifically recognized the C-36 peptide of alpha-1-antitrypsin (AAT) and there was no cross-reactivity against various forms including native, polymerized, oxidized, and complex with elastase, of the full length of AAT protein as well as other peptides (Y105, C-10, and C-21) of AAT protein.  The author also attempted to detect intracellular C-36 peptide in human neutrophils and PBMCs by fluorescent immunocytochemistry and flow cytometry using the antibody.  Overall, the novel antibody will be a useful tool for investigation of the C-36 peptide in in vitro and in vivo experiments, and this paper will be a valuable contribution to the field after revision considering the comments below:

Major comments

1) Quantitative analysis is a useful approach for investigating whether the C-36 peptide is expressed under basal condition and extracellular stimuli such as LPS treatment may affect the intracellular C-36 expression.  To examine the effects of LPS treatment on the C-36 contents in human neutrophils and human PBMCs, quantitative ELISA using the novel monoclonal antibody with synthetic C-36 peptide as a standard should be done.

2) Although the authors indicate that Double labelling experiment showed very limited co-localization between C-36 (red, stained with C42 mAb) and full-length AAT (green, stained with polyclonal anti-AAT) patterns in untreated (upper raw) and LPS-stimulated (lower raw) neutrophils (Page 6 Lines 171-174), for this Reviewer, the fluorescence detected by anti-AAT antibody looks similar to that detected by anti-C-36 peptide antibody especially in the untreated cells (Figure 4, upper).  Furthermore, although the NETs structure released from neutrophils was detected by DAPI staining in Figure 3B, the NETs structure was not observed in Figure 4 (lower), suggesting that these cells were in the different stage of NETosis.  For carefully analyzing the co-localization of the C-36 peptide and the full-length AAT protein in human neutrophils with or without LPS treatment, the author should reperform the same experiments to obtain distinctive NETs and analysis of the orthogonal sections of 3D scanned specimen about the fluorescent images.

Minor comments

1) There is no descriptions about a series of panels shown in the middle of Figure 6A and B? Please explain.

2) Although the authors mentioned "Suppl. figure 1" (Page 4, Line 128), this Reviewer don’t find the figure.  Please correct.

3) There are some typos (e.g., Page 6 Line 179) and some abbreviation, the full name of which is not mentioned in the manuscript (e.g., PBMCs).  Please correct.

Author Response

We thank you very much for the evaluation of our manuscript and for the valuable comments and suggestions. Please find responses in the attachment.

Reviewer 2 Report

There paper is of value for those looking into the peptide and its effects. There are however some pressing scientific questions that need to be addressed

Minor

Abstract - italicize in vitro and in vivo. 

Line 135 - degrees C. 

Line 137 TBS + (pace needed) and also incubated overnight instead of overnight incubated. 

Scheme 1? Maybe call it figure. 

Please insert markers with gels. 

Line 179 - degrees missing 

Line 201, Figure 6A, not common to see 6,A. 

Line 292 degrees C 

Line 400ish and others - 106 cells without the 6 superscripted. Please check throuout the manuscript. And in the sentence, should not be bracketed. 

Major 

It seems important to epitope map by synthesizing short and varying segments of the peptide to determine the binding region. It is curious why denaturing at 44 aa peptide would cause the epitope to be lost. This suggests that the epitope is not sequence specfic but has something to do with structural aspects which may not make it specific. 

Permeabilization was not mentioned until M&Ms, I suggest stating them upfront in the captions and in the results. 

Please show experiments or staining determing neutrophil purity, even if in supplementary. 

While PBMCs are used good to discuss what these cells are likely since neutrophils are dominant WBCs anyway. 

Controls of permeabilization not shown? To include in supplementary. 

Author Response

(The authors gave the same response as above.)

Round 2

Reviewer 1 Report

The authors sufficiently responded to all the reviewers' concerns.

Author Response

Thanks for your comments

Reviewer 2 Report

While I am sympathetic that perform epitope mapping experiments are tedious on their own, in this context it is not difficult to order different segments of the 36 aa peptide given that it is curious how upon denaturation the epitope can be lost. This can be somewhat glossed over if characterization data are tight  e.g. SPR(or equivalent) instead of dot blots.

Just a comment that for peptide purity, 78% is not really high purity. Given this is a paper reporting a novel antibody, perhaps there should be data showing that the antibody is not binding to the carrier protein (which is not described in M&M ).

There is also no characterization data of the C43 mAb being IgM?

The above does need somewhat some addressing if not all. Since basic characterization data of IgM, and ensuring that the antibodies are not reacting to the carrier protein instead, should be somewhat available, they can be placed in supplementary.

Minor- suggest the authors proofread carefully given this is a revised version

Line 86 - "lowers amount" -

Line 135 - no degrees

Line 410- 2x106
